# Age- and Maturity-Related Variations in Morphology, Body Composition, and Motor Fitness among Young Female Tennis Players

**DOI:** 10.3390/ijerph16132412

**Published:** 2019-07-07

**Authors:** Mustafa Söğüt, Leonardo G.O. Luz, Ömer Barış Kaya, Kübra Altunsoy, Ali Ahmet Doğan, Sadettin Kirazci, Filipe Manuel Clemente, Pantelis Theodoros Nikolaidis, Thomas Rosemann, Beat Knechtle

**Affiliations:** 1Faculty of Sport Sciences, Kırıkkale University, Kırıkkale 71000, Turkey; 2LACAPS, Federal University of Alagoas, 57300-000 Arapiraca, Brazil; 3CIDAF, University of Coimbra, 3004-531 Coimbra, Portugal; 4Department of Physical Education and Sports, Middle East Technical University, Ankara 06010, Turkey; 5Polytechnic Institute of Viana do Castelo, School of Sport and Leisure, 4960-320 Melgaço, Portugal; 6Instituto de Telecomunicações, 6200-001 Delegação da Covilhã, Portugal; 7Exercise Physiology Laboratory, 18450 Nikaia, Greece; 8Institute of Primary Care, University of Zurich, 8091 Zurich, Switzerland; 9Medbase St. Gallen Am Vadianplatz, 9001 St. Gallen, Switzerland

**Keywords:** anthropometric measurements, biological maturation, fitness, tennis

## Abstract

The purpose of this study was to examine the effects of age and maturity on anthropometric and various fitness characteristics in young competitive female tennis players. Sixty-one players, aged 10.4–13.2 years (11.8 ± 0.8) were measured for standing and sitting heights, body mass, skinfolds, grip strength, and agility, and dichotomized into two age (U12 and U14) and maturity (earliest and latest) groups according to their chronological age and maturity status. The results revealed significant age effects for stature, sitting height, leg length, and hand grip in favor of the older players. Girls contrasting in maturation differed significantly for all anthropometric and physical performance variables except for body mass index (BMI), body fat percentage (BF%), and hexagon agility test. The earliest maturing group showed significantly higher values for anthropometric measures and better results in the hand grip test than the latest maturing group. After controlling for chronological age, differences were revealed between contrasting maturity groups in stature, sitting height, BF%, and the hand grip test. The findings highlight the age- and maturity-related trends in body size and muscular strength among young female tennis players in the pubertal period. Nevertheless, the differences in the body composition and agility of the contrasting age and maturity groups were negligible.

## 1. Introduction

National and international sports competitions for children and youth are generally structured in age groups based on chronological age [1,2,3]. Children born in the same calendar year, on the other hand, may vary considerably in their morphological and motor performance characteristics [4,5,6]. These variations are vastly attributed to the discrepancies in biological maturity status [7]. 

Previous studies have documented the effects of maturity status on the physical and physiological traits of young male athletes in various sports such as basketball [8,9,10], handball [11], ice hockey [6], rugby [12], and soccer [13,14,15]. A great majority of the findings of the abovementioned investigations highlight the superiority of early maturing children on measured parameters when compared to their late maturing counterparts. Particularly in team sports requiring power, speed, and strength, a considerable decrease occurs between the numbers of late maturing boys aged 11–12 and those aged 15–16 years [16]. Similarly, positive influences of advanced maturity in upper and lower power, hand grip strength, and speed were noted for young male tennis players [17].

Inter-individual differences in maturity during the transition period from childhood to adolescence are more obvious in girls [18]. In non-athletic cohorts, girls advanced in maturation were found to display higher levels of overweight, obesity prevalence and sedentary behavior when compared to their less mature counterparts [19]. Furthermore, no significant maturity-group differences were observed by Freitas et al. [20] in cardiorespiratory fitness, muscular strength, and flexibility. At the elite level, being advanced or late in maturation may play a key role in the context of talent identification. For example, while the late maturing young female athletes constitute the majority in gymnastics, they are under-represented in sports characterized by body size and muscular strength, such as swimming and tennis [1,16]. Supportively, results of the recent studies indicated that age at peak height velocity [21] and percentage of the predicted adult stature [22] were significantly associated with the actual performance (ranking) of young female tennis players. 

Nonetheless, the current literature provides limited evidence on the disparities among youth female tennis players with contrasting maturities. To the best of the authors’ knowledge, there are two published reports that examined the effects maturity-associated differences on the performance of competitive young tennis players [17,23]. Their results, however, demonstrated inconsistency in several motor performance tasks, for example, in strength, agility, and aerobic endurance. Therefore, the purpose of this study was to determine the effects of age- and maturity-related variations on anthropometric, muscular strength, and agility characteristics among young competitive female tennis players. Considering the available data, it was hypothesized that older and more mature girls would show higher values in anthropometric and strength measures. A comparable result was expected between age and maturity groups in agility.

## 2. Materials and Methods

### 2.1. Participants

The sample included sixty-one nationally ranked Turkish young female tennis players, aged 10.4–13.2 years (11.8 ± 0.8 years). The inclusion criteria for the participants were: minimum of two years of experience in regular tennis training, one year of participating in national tournaments, and at least 4 h/week of tennis practice.

### 2.2. Procedures

The measurement protocols and purposes of the study were presented to all the players and their parents. Written informed consent was obtained in accordance with the Declaration of Helsinki. Ethical approval was obtained from the Human Subjects Ethics Committee of Middle East Technical University (approval number of the study is 281-ODTÜ-2019). Anthropometric measurements were performed by a single observer according to the reference manual [24]. Fitness assessments were administered in an indoor tennis court. Players were requested to complete a 20 min warm-up procedure, which included ten minutes of jogging and ten minutes of stretching. The duration of the recovery time between the performance measurements was set to three minutes.

Competitions for junior tennis players were organized into age categories (U12, U14, etc.). Therefore, participants were divided into two age (U12 and U14) groups. Moreover, they were distributed into two maturity (earliest and latest) groups according to their estimated maturity status.

### 2.3. Measurements

#### 2.3.1. Anthropometric Measurements

Five anthropometric variables (stature, sitting height, body mass, and two skinfolds) were measured. Stature and sitting height were measured with a portable stadiometer (Seca 213, Hamburg, Germany) to the nearest 0.1 cm. Body mass was evaluated with a digital weighing scale calibrated to the nearest 0.1 kg. Sitting height was measured with the participants seated on a flat box. The distance between the vertex and the seating surface was recorded. Leg length was obtained by subtracting sitting height from standing height. Triceps (at the marked midpoint between the acromion and olecranon processes) and medial calf (at the largest calf circumference) skinfolds were assessed using a calliper (Holtain Ltd, Crymych, UK) to the nearest 0.2 mm. Body mass index (BMI) was calculated by dividing the weight (kg) by the squared height (m). Percentage of body fat (BF%) was estimated using the equations of Slaughter et al. [25].

#### 2.3.2. Somatic Maturation

Adult stature was predicted with sex-specific equations developed for American children from the Fels Longitudinal Study conducted in South Central Ohio in the United States [24]. Current stature of the child was then expressed as a percentage of the predicted adult stature (%PAS) and was used as an indicator of maturity status at the time of observation. Predictors included chronological age, current stature and body mass from each child and also required the average self-reported stature of the child’s biological mother and father. Two girls of the same chronological age may have the same stature, but one could be closer to adult stature than the other, and the former could be more advanced in maturity status compared to the other [26]. The Khamis–Roche method has been employed as an estimate of biological maturity status in several studies [27,28]. In the current study, girls were grouped using a sample median z-score of the attained %PAS: earliest maturing (*p* > 50%) and latest maturing (*p* < 50%).

#### 2.3.3. Grip Strength

The maximal isometric grip strength of both dominant and non-dominant hands was evaluated using a digital hand dynamometer (T.K.K.5401 Grip-D, Takei, Japan). The players were asked to stay in the standing position and keep their arms straight along with their bodies. The grip span was adjusted according to the player’s hand size. They were requested to squeeze the dynamometer as hard as they could for three seconds. The maximum score obtained from the three trials was used for the data analysis.

#### 2.3.4. Agility

The hexagon agility test was administered to assess the agility performance of the players in accordance with the procedure described in the earlier studies [17,29]. The hexagon shape was drawn measuring 60 cm per side and 120-degree angles between sides. Players faced forward with feet together throughout the measurement. They were asked to jump over each side of the hexagon and return in it again in a clockwise manner. The time to complete three rotations was recorded to the nearest 0.1 seconds using a stopwatch. The assessment was repeated twice and the fastest score was recorded.

### 2.4. Statistical Analysis

Descriptive statistics were calculated for each chronological age group for all variables (Table 1). Kolmogorov–Smirnov was used to test normality, and appropriate log transformations (log 10) were adopted to normalize distributions and meet the assumptions of subsequent analyses. Further, the independent and combined effects of chronological age and maturity status on the anthropometry profile and the physical fitness tests were examined using a multivariate analysis of variance (MANOVA) and a multivariate analysis of covariance (MANCOVA with somatic maturation given by z-scores of attained predicted adult stature as covariate), separately for anthropometry and physical fitness domains. When MANOVA or MANCOVA detected a statistically significant effect, subsequent analysis of variance (ANOVA) and analysis of covariance (ANCOVA) were used to detect the contribution of the single dependent variables to the multivariate solution (Table 2). Descriptive statistics were calculated for each maturity status group. Then, the groups were tested using an independent sample t-test, and the standardized differences between the means were reported using Cohen’s d values, interpreted as follows: <0.20 (trivial), 0.20–0.59 (small), 0.60–1.19 (moderate), 1.20–1.99 (large), 2.0–3.9 (very large), and >4.0 (extremely large) [30]. Finally, ANCOVA was used to examine maturity status differences after controlling for chronological age in the anthropometric characteristics and the physical fitness tests (Table 3). Data were analyzed using IBM SPSS 22.0 (SPSS, Inc., Chicago, IL, USA). The significance level was set to 5% for all inferential statistics.

## 3. Results

Descriptive statistics of the chronological variables, the anthropometric profile, and the physical fitness tests by age groups are illustrated in Table 1. Older girls showed higher values for attained predicted adult stature and all anthropometric variables than their younger peers. Also, for all the physical performance characteristics, differences were found between the groups, with the older girls presenting better results.

Table 2 summarizes the MANOVAs for all the anthropometric characteristics (Wilks’ lambda = 0.709, F = 4.521, *p* < 0.01) and the physical fitness tests (Wilks’ lambda = 0.843, F = 3.543, *p* < 0.05) and shows a significant chronological age effect in both domains. In addition, subsequent ANOVAs were produced separately for each measure and test, and differences were significant for stature (F = 17.640, *p* < 0.01), sitting height (F = 11.202, *p* < 0.01), leg length (F = 18.762, *p* < 0.01), dominant hand grip (F = 5.917, *p* < 0.05) and non-dominant hand grip (F = 7.111, *p* < 0.05). The results of the MANCOVAs used to examine the chronological age differences (with somatic maturation as the covariate) are also presented in Table 2. The MANCOVAs showed a significant age effect on the anthropometric domain (Wilks’ lambda = 0.746, F = 3.671, *p* < 0.01), specifically for stature (F = 10.125, *p* < 0.01), sitting height (F = 16.877, *p* < 0.01), body mass (F = 9.107, *p* < 0.01) and BMI (F = 6.070, *p* < 0.05). For the physical fitness domain, no age effect emerged when adjusted for maturation.

The descriptive statistics, results from independent sample t-test, and ANCOVAs (controlling for chronological age) of the anthropometric profile and the physical fitness comparison between the two contrasting maturity groups are presented in Table 3. Girls contrasting in maturation differed significantly for all anthropometric and all physical performance variables except for BMI, BF% and hexagon agility test. The earliest maturing group showed significantly higher values for the anthropometric measures and better results in the hand grip tests than the latest maturing group. After controlling for chronological age, differences could be revealed between the contrasting maturity groups in stature (F = 5.719, *p* < 0.05), sitting height (F = 5.226, *p* < 0.05), BF% (F = 4.830, *p* < 0.05) and the dominant hand grip test (F = 4.253, *p* < 0.05).

## 4. Discussion

The aim of this cross-sectional study was to examine the age- and maturity-associated variations in the morphology, body composition, and motor fitness among young female tennis players. The results demonstrate that older and earliest maturing girls outperformed their younger and latest maturing counterparts in terms of grip strength. Similar agility scores were found between the different maturity and age groups. Unfortunately, there is a dearth of evidence on the maturity-associated variations in the physical fitness characteristics of young female athletes [16]. Nevertheless, the results are partly in line with the findings of former examinations [17,23] which investigated the impacts of biological maturity on the physical fitness attributes among young female tennis players. While comparable values in regard to grip strength performances were found between girls in different maturity statuses by Van Den Berg, Coetzee, and Pienaar [23], significantly greater values were noted in favor of girls advanced in maturity by Myburgh et al. [17]. There is a general agreement that children’s muscular strength is associated with their age, biological maturation, stature, and body mass [31,32,33]. According to Malina, Bouchard and Bar-Or [7], there is a linear increase in the grip strength performances of girls between the ages of 11 and 17, and the superiority of early maturing girls is due to their larger body size. Contrarily, in accordance with the results of the present study, the findings of the aforementioned studies showed consistency when considering the agility performances. The paucity of data on the contributions of maturation on agility prohibits explaining this finding. However, Malina, Bouchard, and Bar-Or [7] stated that the agility performance of girls improves substantially between the ages of 5 and 8 and remains constant after the ages of 13 and 14.

The findings regarding the main anthropometric variables indicate significant differences in favor of the older and earliest maturing players. These advantages were more specific for standing and sitting height when both age and maturity status were controlled as covariates. This finding can be explained by the fact that, when compared to their less mature peers, early maturing girls usually reach their peak height velocity one year earlier [7]. During the pubertal growth spurt, girls attain their peak height velocity between 10 and 13 years of age [34] and generally at about 12 years [7]. Consistent results were also obtained by previous investigations for tennis players [21,22,35].

The fat percentage and BMI of different age and maturity groups showed comparable values. These results are in accordance with the findings of earlier studies [36,37,38] that show no significant maturity-related differences in the adiposity parameters among girls. This may be due to the possible positive influences of regular tennis training on the maintenance of BF% and BMI in consecutive ages. According to Malina and Geithner [39], body composition is sensitive to organized training. In a comparative study, lower total and regional BF% values were reported for young tennis players playing tennis twice a week, when compared to their age- and maturity-matched non-active controls [40]. Similar results were also obtained for young female gymnasts [41]. The role of various exercise interventions on decreasing the BF% of children and adolescents is well documented [42,43,44].

## 5. Conclusions

The purpose of the present study was to examine the effects of age and biological maturation on the anthropometric and various fitness characteristics in young competitive female tennis players. The results reveal maturity-related differences in favor of the earliest maturing girls for the main anthropometric variables and muscle strength. Nonetheless, the differences between the maturity groups for body composition and agility were insignificant. These results suggest that the maturity status of young female tennis players must be considered within the context of talent identification in order to value their body size and strength adequately. The present study has several limitations. First, the cross-sectional observational design employed in this study precludes cause and effect interpretations. Second, the relatively small sample size prohibits generalization of the findings. Third, the study was limited to two physical fitness tasks. Fourth, tennis-specific skills were not examined. Future studies are recommended to extend this observation in a large sample of young female athletes.

## Figures and Tables

**Table 1 ijerph-16-02412-t001:** Descriptive statistics (minimum, maximum, mean and standard deviation) of the chronological variables, anthropometric profile and physical fitness tests by age groups.

Variables	Age Groups							
	U12 (*n* = 36)				U14 (*n* = 25)			
	Min	Max	Mean	SD	Min	Max	Mean	SD
Chronological age (years)	10.4	11.9	11.2	0.4	12.0	13.2	12.6	0.4
Percentage of PAS (%)	78.0	88.0	82.9	2.4	81.0	91.0	88.6	2.4
Stature (cm)	137.1	165.5	149.9	7.2	138.4	167.5	157.7	6.9
Sitting height (cm)	72.5	89.8	78.7	4.1	72.2	88.4	82.4	4.3
Leg length (cm)	64.6	78.3	71.2	3.7	66.2	82.0	75.3	3.4
Body mass (kg)	31.9	64.9	42.2	8.3	31.1	76.0	46.9	10.0
Body mass index (kg/m^2^)	15.4	25.5	18.6	2.3	8.6	26.9	18.4	3.4
Body fat percentage (%)	13.3	39.5	22.4	5.7	14.0	38.3	21.7	5.2
Dominant hand grip (kg)	14.1	32.0	19.7	3.5	14.9	30.1	22.2	4.3
Non-dominant hand grip (kg)	12.7	31.0	17.4	3.5	14.6	28.9	19.7	3.6
Hexagon agility test (s)	11.0	17.5	13.7	1.5	10.8	17.1	13.0	1.4

U12 (under 12.0 years); U14 (under 14.0 years); PAS (predicted adult stature).

**Table 2 ijerph-16-02412-t002:** Multivariate analyses of variance (MANOVA) and multivariate analyses of covariance (MANCOVA) with somatic maturation given by z-scores of the attained predicted adult stature as the covariate) to examine the effects of chronological age on anthropometry and physical fitness.

	MANOVA					MANCOVA				
Dependent Variables	Chronological Age					Chronological Age ^a^				
	Test	λ de Wilks	*F*	*p*	*η^2^_p_*	Test	λ de Wilks	*F*	*p*	*η^2^_p_*
Anthropometry	MANOVA	0.709	4.521	<0.01		MANCOVA	0.746	3.671	<0.01	
Stature	ANOVA		17.640	<0.01	0.230	ANCOVA		10.125	<0.01	0.149
Sitting height	ANOVA		11.202	<0.01	0.160	ANCOVA		16.877	<0.01	0.225
Leg length	ANOVA		18.762	<0.01	0.241	ANCOVA		1.353	0.250	0.023
Body mass	ANOVA		3.911	0.053	0.062	ANCOVA		9.107	<0.01	0.136
Body mass index	ANOVA		0.148	0.702	0.002	ANCOVA		6.070	<0.05	0.095
Body fat percentage	ANOVA		0.237	0.628	0.004	ANCOVA		1.946	0.168	0.032
Physical fitness	MANOVA	0.843	3.543	<0.05		MANCOVA	0.949	1.010	0.395	
Dominant hand grip	ANOVA		5.917	<0.05	0.108	ANCOVA		3.000	0.089	0.049
Non-dominant hand grip	ANOVA		7.111	<0.05	0.091	ANCOVA		2.055	0.157	0.034
Hexagon agility test	ANOVA		2.944	0.091	0.048	ANCOVA		0.437	0.511	0.007

*η^2^_p_* (partial eta square); ^a^ somatic maturation given by z-scores of attained predicted adult stature as covariate.

**Table 3 ijerph-16-02412-t003:** Descriptive statistics (mean and standard deviation), results from independent sample t-test, effect size given by Cohen’s *d* and ANCOVA (controlling for chronological age) of the anthropometric profile and physical fitness comparison between the two contrasting maturity groups.

	Contrasting Maturity Group				Independent Sample *t*-test		Effect Size		ANCOVA		
	Latest Maturing (*n* = 31)		Earliest Maturing (*n* = 30)								
Dependent Variables	Mean	SD	Mean	SD	*t* *(df = 59)*	*p*	*d*	Qualitative	*F*	*p*	*η^2^_p_*
Chronological age (years)	11.1	0.4	12.5	0.5	−12.154	<0.001	−3.05	Very large	-	-	-
Stature (cm)	148.8	6.6	157.5	6.8	−5.042	<0.001	−1.28	Large	5.719	<0.05	0.090
Sitting height (cm)	78.1	3.6	82.4	4.3	−4.238	<0.001	−1.07	Moderate	5.226	<0.05	0.083
Leg length (cm)	70.7	3.7	75.1	3.3	−4.840	<0.001	−1.23	Large	3.771	0.057	0.061
Body mass (kg)	40.6	6.4	47.8	10.4	−3.257	<0.01	−0.82	Moderate	0.078	0.781	0.001
Body mass index (kg/m^2^)	18.2	2.0	18.8	3.5	−0.774	0.443	−0.21	Small	0.650	0.423	0.011
Body fat percentage (%)	21.7	4.7	22.6	6.2	−0.620	0.538	−0.16	Trivial	4.830	<0.05	0.077
Dominant hand grip (kg)	19.2	3.0	22.3	4.4	−3.192	<0.01	−0.81	Moderate	4.253	<0.05	0.068
Non-dominant hand grip (kg)	16.9	2.6	19.9	4.1	−3.410	<0.01	−0.86	Moderate	1.303	0.258	0.022
Hexagon agility test (s)	13.7	1.6	13.0	1.4	1.722	0.090	0.46	Small	0.054	0.817	0.001

*η^2^_p_* (partial eta square).

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
