# Peer review of "Age- and Maturity-Related Variations in Morphology, Body Composition, and Motor Fitness among Young Female Tennis Players"

_ijerph, 2019, doi:10.3390/ijerph16132412_

Round 1
Reviewer 1 Report
Introduction
The introduction is adequate, although there are only two bibliographical citations from the last three years. Isn't there anything relevant that can be included? It is recommended to make a review of the scientific literature including more current citations that justify and verify the importance and problem of the study.
Method
For such a journal, the study sample is small. It is recommended to increase the study sample so that this research can be generalized.
In the participants section, the contextualization of the study subjects must be included. In addition, lines 69-72, it is more correct to appear in the study procedure.
Results
Adequate results, although in a simple way, could also be to include lineal and/or Pearson correlations in the results in order to establish relationships between the main variables.
Why do table 2 and 3 appear in the discussion? Download the discussion paragraph and adjust the criterial.
Discussion and conclusions
In the first paragraph of the discussion it would be interesting to discuss the objective of the study with research that has carried out the same procedures or is of a similar nature.
I think there are more study limitations than the authors show, such as the study sample, because it cannot be generalized.
The prospects for the future are good, although it is appropriate to include the practical applications of this research in this area.
In general, it seems to me to be a manuscript with a good structure and normal and publishable statistical analyses. The only two aspects that make it lose quality are the study sample (it is too small to generalize) and the bibliographic references (because although most of them are from this century, there are hardly any studies from the last 3-5 years, which can be a serious problem, since these relationships may have been studied and the population has changed a lot in 10-15 years).
Author Response
Comments and Suggestions for Authors
Introduction
The introduction is adequate, although there are only two bibliographical citations from the last three years. Isn't there anything relevant that can be included? It is recommended to make a review of the scientific literature including more current citations that justify and verify the importance and problem of the study.
Answer: We agree with the expert reviewer and some of the references were updated.
Method
For such a journal, the study sample is small. It is recommended to increase the study sample so that this research can be generalized.
Answer: We agree with the expert reviewer but unfortunately authors are unable to measure more participants due to time constriction and availability of the prospective sample. However, we have included this point raised by the reviewer as a pragmatic suggestion for further research.
In the participants section, the contextualization of the study subjects must be included. In addition, lines 69-72, it is more correct to appear in the study procedure.
Answer: We agree with the expert reviewer and the contextualization of the study subjects was added. Sentences within the line 69 and 72 were placed in procedure section.
Results
Adequate results, although in a simple way, could also be to include lineal and/or Pearson correlations in the results in order to establish relationships between the main variables.
Answer: We agree with the expert reviewer. However, authors kindly like to keep the statistical analysis as submitted.
Why do table 2 and 3 appear in the discussion? Download the discussion paragraph and adjust the criterial.
Answer: We agree with the expert reviewer. This issue might be due to editorial preference. Table 2 and 3 were originally placed in the results section for the submitted manuscript. They are again located in the results.
Discussion and conclusions
In the first paragraph of the discussion it would be interesting to discuss the objective of the study with research that has carried out the same procedures or is of a similar nature.
Answer: We agree with the expert reviewer and the first paragraph of the discussion section was revised.
I think there are more study limitations than the authors show, such as the study sample, because it cannot be generalized.
Answer: We agree with the expert reviewer and limitations of the study were expanded.
The prospects for the future are good, although it is appropriate to include the practical applications of this research in this area.
Answer: We agree with the expert reviewer and practical application of the study was added.
In general, it seems to me to be a manuscript with a good structure and normal and publishable statistical analyses. The only two aspects that make it lose quality are the study sample (it is too small to generalize) and the bibliographic references (because although most of them are from this century, there are hardly any studies from the last 3-5 years, which can be a serious problem, since these relationships may have been studied and the population has changed a lot in 10-15 years).
Answer: We agree with the expert reviewer and we have of course incorporated these points in the relevant sections of the manuscript.
Reviewer 2 Report
Major comments:
- no reference is made to any control group, in this type of research is required,
- please specify the paper contribution and develop the research goal, the analysis of the diversity of anthropometric parameters among older and young girls seems too trivial. The authors in their research have noticed that with age, the anthropometric parameters of girls change. I would be surprised if it was different.
- practical conclusions are missing at work. What is the contribution of such research to the training process of young tennis players?
Author Response
Comments and Suggestions for Authors
Major comments:
- no reference is made to any control group, in this type of research is required,
Answer: We agree with the expert reviewer and information for the non-athletic population was presented in the introduction section through citing previous studies. We have also noted this important point raised by the reviewer as a pragmatic suggestion for further research.
- please specify the paper contribution and develop the research goal, the analysis of the diversity of anthropometric parameters among older and young girls seems too trivial. The authors in their research have noticed that with age, the anthropometric parameters of girls change. I would be surprised if it was different.
Answer: We agree with the expert reviewer and the prospective contribution of the study was added to the conclusion section. In order to establish integrity with the analysis, the differences between age and maturity groups were discuss for all variables.
- practical conclusions are missing at work. What is the contribution of such research to the training process of young tennis players?
Answer: We agree with the expert reviewer and practical conclusion of the study was incorporated in the discussion section.
Reviewer 3 Report
Please see attached pdf file for comments. Thank you for the opportunity to review your work.

Author Response
Comments and Suggestions for Authors
Please see attached pdf file for comments. Thank you for the opportunity to review your work.
Review.
As reported by the authors of the manuscript, many sports, especially at national and
elite levels, group young athletes for competition by chronological age and not on
morphological or functional characteristics. Maturity of young athletes has been described in
various sports and may be used as a predictor and success. However, the authors report that
only two published studies have investigated maturity levels of young women tennis players.
The authors describe several anthropometric measures for sixty-one young (10.4 to
13.2 years old) female tennis players. The measures included standing and sitting height, body
mass, skin fold measures. In addition, two functional measures of and grip strength and a
hexagon agility test were completed. The young women were divided into two age groups of
U12 and U14 and also divided into early or late maturity. The authors report a significant age
effect for height and grip strength. There was an effect of maturation on all variables tested
except calculated BMI and body fat percent. Earlier maturation young women showed higher
values for arthropometric measures and better grip strength scores. Controlling for
chronological age revealed differences in height, percent body fat, and hand grip strength
between maturity groups.
The manuscript is well organized with several in depth tables yet no figures of
representative data. The manuscript is nicely written yet I suggest some word choice or
sentence structure changes. Statistical analysis was detailed and appropriate. Some more
detail is needed for the introduction and specifically the purpose of the study, and more detail
describing the sample population studied. See below for specific comments.
Specific Critique.
Introduction.
Line 61. Re write this word. Researches
“To the best of authors’ knowledge there are two published researches that examined
the maturity-associated differences on the performance of the competitive young tennis
players [20,21]. “
Answer: We agree with the expert reviewer and the word “researches” was replaced with “reports”.
Line 62. Please expand on the ‘demonstrated inconsistency’ of the two previous studies
regarding tennis player maturity.
“Their results, however, demonstrated inconsistency.“
Answer: We agree with the expert reviewer and the sentence was revised.
Line 62 to 64. Is the purpose of the study to then resolve these previous inconsistencies? If so,
please provide more detail for purpose of the study. What are defined as “functional”
capabilities? (see below re: physical fitness)
Answer: We agree with the expert reviewer and the last paragraph of the introduction section was revised. The purpose statement was detailed in regard to reviewer’s comments.
Line 62 to 64. The current purpose statement is a bit wordy. Maybe delete “the” and simply
write “…. anthropometric ….” or …
“was to determine variations in age and maturity related anthropometric …..”
Answer: We agree with the expert reviewer and “the” was deleted.
Materials and Methods.
Line 67 to 69. Can you please expand on the competitive level of these tennis players?
Describe what is considered, defined as “competitive”. Is this a representative sample? Was
their specific inclusion criteria or exclusion criteria? What are official tournaments? How long
have they played tennis? Where were these players recruited? Race of players (needed for
%BF and %PAS).
Answer: We agree with the expert reviewer and the participants section was revised in respect to the reviewer’s comments.
“The sample included sixty-one young competitive female tennis players, aged 10.4-13.2 years
(11.8 ± 0.8 years), participating in regular tennis training and official tournaments. “
Answer: We agree with the expert reviewer and the section was revised in respect to the reviewer’s comments.
Line 83. How was seated height measured? How was leg length measured (anatomical
locations)? Anatomical or reference for locations for skin folds.
Answer: We agree with the expert reviewer. Anatomical locations and the explanation for the measurements were added.
Line 97. Was the height of the biological parents measured? Or was this self-reported. Did the
authors have data for stature for both biological parents? Please add to procedures.
Answer: We agree with the expert reviewer and the phrase “self-reported” was added to the relevant sentence.
Line 100. The Khramis-Roche method (1994) is limited to white American children (from the
state of Ohio). In addition, the study by Malina et al (2005) only included boys (playing
American football) in their estimation of maturity level. Please find reference that are report
the validity of the Khramis-Roche method for the population studied.
Answer: We agree with the expert reviewer. The study sample includes Caucasian Turkish children. The validity of the Khramis-Roche method for the study population has not been established.
Line 101. What is the rational or ‘early vs late’ maturity? Why not divide into three maturity
levels (early, average, late) as per L Van Den Berg et al (2006) and others (Malina et al 2005).
Answer: We agree with the expert reviewer but our sample was not large enough to rigorously classify participants into three maturity groups.
Line 132. I thank the authors for reporting Cohens d values.
Answer: We thank reviewer for this kind comment.
Results.
Line 146 and Table 1. Why divide into U12 and U14 players. Explain these divisions. Are these
specific to tennis? At what chronological age are they considered at these age divisions (e.g.
birthday). Did any U12 players complete at U14 levels? Again, is there a purpose to divide
these players into these two groups.
Answer: We agree with the expert reviewer. Youth tennis competition is divided into age categories based on chronological age as defined by a player’s date of birth. An explanation for this comment was added in the procedure section.
Line 175 and Table 2. Formatting of this table needs to be adjusted. Are Physical Fitness
measures (grip and agility) defined as the functional variables you wish to investigate? Be
consistent throughout the manuscript with terminology (physical fitness vs. functional
capabilities).
Answer: We agree with the expert reviewer and the phrase “functional” was deleted in all parts of manuscript. Physical fitness was used in order to provide consistency.
Discussion.
Line 171. Delete “on the other side”
Answer: We agree with the expert reviewer and the phrase was deleted.
Line 189. Replace ‘contrasting’ with different.
Answer: We agree with the expert reviewer and the word was replaced.
Line 190. Delete “the” earlier studies.
Answer: We agree with the expert reviewer and “the” was deleted.
Line 193. Delete “the” organized training
Answer: We agree with the expert reviewer and “the” was deleted.
Line 194. Delete ‘for example’
Answer: We agree with the expert reviewer and the phrase was deleted.
Line 197. Delete “Supportively”
Answer: We agree with the expert reviewer and the word was deleted.
Line 200. Delete ‘on the other side’
Answer: We agree with the expert reviewer and the phrase was deleted.
Line 201. Again, replace contrasting with ‘different’ or some other word choice.
Answer: We agree with the expert reviewer and the word was replaced.
How do the results of the present study add to the literature? I am not convinced about the
importance of the present study. What the current study adds to the literature should be
discussed.
Answer: We agree with the expert reviewer and practical application of the study was added to the conclusion section.
Conclusions.
Could the authors please list what were the specific conclusions of the study. Or in
simple sentences, what were the conclusions. This section is confusing and needs to be
expanded.
Answer: We agree with the expert reviewer and the conclusion section was expanded.
Line 217. Delete “in conclusion”
Answer: We agree with the expert reviewer and the phrase was deleted.
Line 217 to 218. Re-write the first purpose statement in this conclusion section.
Answer: We agree with the expert reviewer and the purpose statement in the conclusion section was revised.
Line 219. Delete “the” body size. Re write this second sentence as well.
Answer: We agree with the expert reviewer and the sentence was revised.
Line 220. Again, different word choice for ‘contrasting’
Answer: We agree with the expert reviewer and the sentence was revised.
Line 221. Delete “it must be taken into account”
Answer: We agree with the expert reviewer and the phrase was deleted.
Line 221. What is ‘negligible’? Not significant?
Answer: We agree with the expert reviewer and the word “negligible” was replaced with “insignificant.
Line 223. Re write last sentence.
Answer: We agree with the expert reviewer and the sentence was revised.
References.
Reference #29 title should be italics
Answer: We agree with the expert reviewer and the reference was revised.
Reference #4 and #5 need to correct one of the authors names.Submission Date
Answer: We agree with the expert reviewer and the references were revised.
Reviewer 4 Report
Dear Authors,
Attached you can find my review report.
Best regards,
Reviewer

Author Response
Comments and Suggestions for Authors
Dear Authors,
Attached you can find my review report.
Best regards,
Reviewer
Age and maturity-related variations in morphology, body composition, and motor fitness among young female tennis players
Review
Thank you for the opportunity to review this manuscript. The present study aimed to examine the effects of age and maturity on the anthropometric and functional characteristics in young competitive female tennis players. The manuscript covers interesting literature, however, requires a better structure and more clarity. I have provided examples below.
Introduction.
1. The introduction section needs to be explained in more depth. More scientific studies in relation to the analyzed problem should be provided and discussed regarding previously published articles. The explanation of specific age groups (early, middle, late adolescence age) and specifics of it would be beneficial. The pubertal growth aspects could be included and explained in more detail.
Answer: We agree with the expert reviewer and the introduction section was revised in regard to reviewer’s comments.
2. The theoretical framework is not discussed in the introduction section. What is a theoretical framework of your study?
Answer: We agree with the expert reviewer and the introduction section was revised.
3. At the end of the introduction, a clearly defined research question(s) of the study could be included.
Answer: We agree with the expert reviewer and hypotheses were set.
4. Did you have any hypotheses for your analyses?
Answer: We agree with the expert reviewer and hypotheses were set.
Methods.
1. Line 24: „The purpose of this study was to examine the effects of age and maturity on the anthropometric and functional characteristics in young competitive female tennis players“. Because of this formulated study purpose, the natural question arises: What is competitive tennis players mean? Can you explain in the methods section? How we should understand the competitive players? How did you recruit them? Based on what explanation the competitive players were chosen? I will give an example: According to Smith (2009) competitive players were chosen based on these criteria: 3 years of participating in competitions., athletes played tennis for no less than 2 years, etc. This information is necessary for analyzing such a topic.
Answer: We agree with the expert reviewer and the participants section was revised.
Line 68 says that „participating in regular tennis training and official tournaments“ So in this sense, if I participate in regular training and started tennis 3 months ago, and participated in one competition - I will be a competitive female tennis player? Hope this example will help to unpack this question. And another question, what kind of tournaments? National, international? It is important for a reader to know.
Answer: We agree with the expert reviewer and the participants section was revised.
Results.
1. Results are well presented.
Discussion.
1. Why table 2 and table 3 appears in the discussion part?
Answer: We agree with the expert reviewer. This issue might be due to editorial preference. Table 2 and 3 were originally placed in the results section for the submitted manuscript. They are again located in the results.
2. Can you add some practical implications at the end of the discussion?
Answer: We agree with the expert reviewer and practical application of the study was added.
Round 2
Reviewer 2 Report
All comments are addressed, I believe it can be accepted at current form.